

# The Berkeley Earth Land/Ocean Temperature Record

Robert A. Rohde[1], Zeke Hausfather[1,2]

[1] Berkeley Earth, Berkeley, CA 94705, USA
[2] Breakthrough Institute, Oakland, CA 94612, USA

*Correspondence to*: Robert A. Rohde (robert@berkeleyearth.org)

**Abstract.** A global land/ocean temperature record has been created by combining the Berkeley Earth monthly land temperature field with spatially-kriged version of the HadSST3 dataset. This combined product spans the period from 1850 to present and covers the majority of the Earth's surface: approximately 57% in 1850, 75% in 1880, 95% in 1960, and 99.9% by 2015. It includes average temperatures in 1°x1° lat/lon grid cells for each month when available. It agrees quite well with records from

Hadley's HadCRUT4, NASA's GISTEMP, NOAA's GlobalTemp, and Cowtan and Way, but provides a more spatially complete and homogeneous temperature field. Two versions of the record are provided treating areas with sea ice cover as either air temperature over sea ice or sea surface temperature under sea ice. The choice of how to assess the temperature of areas with sea ice coverage has a notable impact on global anomalies over past decades due to rapid warming of air temperatures in the Arctic. Accounting for rapid warming of Arctic air suggests ~0.1 ºC additional global-average temperature rise since the 19[th] century

than temperature series that do not capture the changes in the Arctic. Updated versions of this dataset will be presented each month at the Berkeley Earth website (http://berkeleyearth.org/data/), and a convenience copy of the version discussed in this paper has been archived and is freely available at https://doi.org/10.5281/zenodo.3634713 (Rohde & Hausfather, 2020).

## 1 Introduction

Global land-ocean temperature indices combining 2-meter surface air temperature over land with sea surface temperatures (SST)

over oceans are commonly used to assess changes in the Earth's climate. While it is a less physically meaningful metric than earth system total heat content, it is well-measured with reliable data extending back to *c.*1850 for oceans (Kennedy et al., 2011) and as far back as *c.*1750 for land (Rohde et al., 2013a) and is the part of the Earth system most relevant for impacts on human civilization. Sea surface temperatures are used in lieu of marine air temperatures due to scarcity and inhomogeneity of marine air temperature data (Kent et al., 2013), though it is only an imperfect proxy and may be subject to slightly different warming rates

(Cowtan et al., 2015).

A number of prior groups have developed global land/ocean surface temperature indexes, including NASA's GISTEMP (Hansen et al., 2010; Lenssen et al 2019), Hadley's HadCRUT4 (Morice et al., 2012), NOAA's GlobalTemp (Smith et al., 2008; Vose et al., 2012), and the Japan Meteorological Agency (JMA) (Ishihara 2006). Additionally, Cowtan and Way (2014) provide a

spatially-interpolated variant of HadCRUT4 featuring greater spatial coverage. These series differ in a number of respects. They all largely utilize the same set SST measurements drawn from the ICOADS database (Woodruff et al., 2011) and most of the same land temperature records contained in the Global Historical Climatological Network (GHCN) (Lawrimore et al., 2011), though both GISTEMP and HadCRUT4 (and by extension Cowtan and Way) include a modest number of additional land stations, most notably in Antarctica in the case of GISTEMP.




Both GISTEMP and GlobalTemp utilize NOAA's pairwise homogenization algorithm to detect and correct inhomogenities such as station moves or instrument changes in land stations (Menne and Williams 2009), though NASA applies an additional satellite nightlight-based urbanity correction (Hansen et al., 2010). GISTEMP and GlobalTemp both use NOAA's Extended Reconstruction Sea Surface Temperature (ERSST) version 4 (Huang et al., 2014) for SSTs, HadCRUT4 and Cowtan and Way

use HadSST3 (Kennedy et al., 2011), and JMA uses COBE-SST (Ishii et al., 2005). HadCRUT4, GlobalTemp, and JMA include no spatial interpolation outside of 5-by-5 latitude/longitude gridcells, while GISTEMP and Cowtan and Way spatially interpolate temperatures out to regions with no direct station coverage (GISTEMP using a simple linear interpolation technique, while Cowtan and Way uses Kriging).

Here we describe the global land/ocean surface temperature product from Berkeley Earth that combines the Berkeley Earth land temperature data (Rohde et al 2013a; Rohde et al 2013b) with SST data from HadSST3 (Kennedy et al., 2011). It uses a Kriging-based spatial interpolation to provide the greatest possible spatial coverage for the period from 1850 to present. The land data utilizes significantly more land station data (over 40,000 stations) compared to the ~10,000 land stations used by some of the other groups. The land component also includes the novel homogenization technique of the Berkeley Earth temperature record

that detects breakpoints through neighbor difference series comparisons, cuts land stations into fragmentary records at breakpoints, and combines these fragmentary records into a temperature field. The ocean component of the land/ocean product uses an interpolated variant of HadSST v3, whose construction is described below. A version of this dataset has been publically available for some time, but has not been formally described.

## 2 Methods

The Berkeley Earth Land/Ocean temperature record combines the Berkeley Earth land record (Rohde et al 2013a) with SST data from HadSST3 (Kennedy et al. 2011a, Kennedy et al. 2011b). The HadSST3 data is adjusted in several ways. The primary manipulation is to replace the gridded data with an interpolated field using a Kriging-based approach. The HadSST3 data set provides grid cell averages on a 5° by 5° grid and only reports monthly averages for cells where data was present during the month in question. HadSST3 often reports no data for ~40% of ocean grid cells. As described below, the interpolation produces

a more complete field and reduces the component of uncertainty associated with incomplete coverage. While providing a more complete field, the interpolation does not materially change the apparent rate of warming in the oceans.

After interpolation, the ocean temperature anomaly field is merged with Berkeley Earth land anomaly field using the fraction of land / water in each grid cell (typically reported with a 1° by 1° latitude/longitude resolution). As described below, two versions

are considered with respect to the role of sea ice.

### 2.1 Interpolation Method

The HadSST3 gridded fields provide several critical components, the temperature anomaly, the number of observations, and several estimates of the uncertainty (Kennedy et al. 2011a, Kennedy et al. 2011b). The grid cell uncertainties and observation

counts allow one to treat some grid cells as having greater confidence than others. Unlike land surface station data, where each monthly average represents many temperature observations, the ocean observation counts are a true measure of the number of instantaneous SST measurements.





Analogous to Rohde et al 2013a, the core of the interpolation approach is to generate a Kriging-based field using an assumed
distance-based correlation function. As with Rohde et al 2013a, a correlation-based approach is used rather than the more
common covariance-based approach to simplify the computational considerations, and should be adequate as long as the
variance changes relatively slowly with changes in position. A review of both the HadSST data and climate model outputs
suggested that the temperature to distance correlation function could be modeled effectively via the same spherical correlation
function approach used for land surface temperatures:


$$R(d) = R_0 \left(1 - \frac{d}{d_{max}}\right)^2 \left(1 + \frac{d}{2\, d_{max}}\right), \ d < d_{max} \tag{1}$$

$$R(d) = 0, \ d \geq d_{max}$$

The empirically estimated distance parameter $d_{max}$ was found to have a value of 2,680 km based on the spatial variance of the
HadSST monthly averages. This is similar to, though somewhat smaller than, the 3,310 km scale adopted in the land surface
temperature study (Rohde et al. 2013a). By contrast, the local correlation parameter $R_0 = 0.47$ was estimated to be much lower
in the oceans (compared to 0.86 on land). This is due to two factors. Firstly, ocean observations are individual measurements
whereas land observations reflect monthly averages. Secondly, the typical monthly fluctuations in the oceanic environment are
much smaller in than on land, causing a reduced signal-to-noise ratio. The estimation of $R_0$ was based on a comparison of the
variance in HadSST grid cells with a single measurement to those with > 100 observations. The latter condition provides a proxy
for cells where the random portion of measurement and sampling uncertainty could plausibly be neglected.

Figure 1 shows an empirically estimated average correlation versus distance between HadSST grid cells. This shows the
empirical length scale, though a larger intercept is used (~0.75) reflecting the fact that the average HadSST grid cell incorporates
many observations. The lower value for $R_0$ represents the typical relationship between a single measurement and the monthly
average.

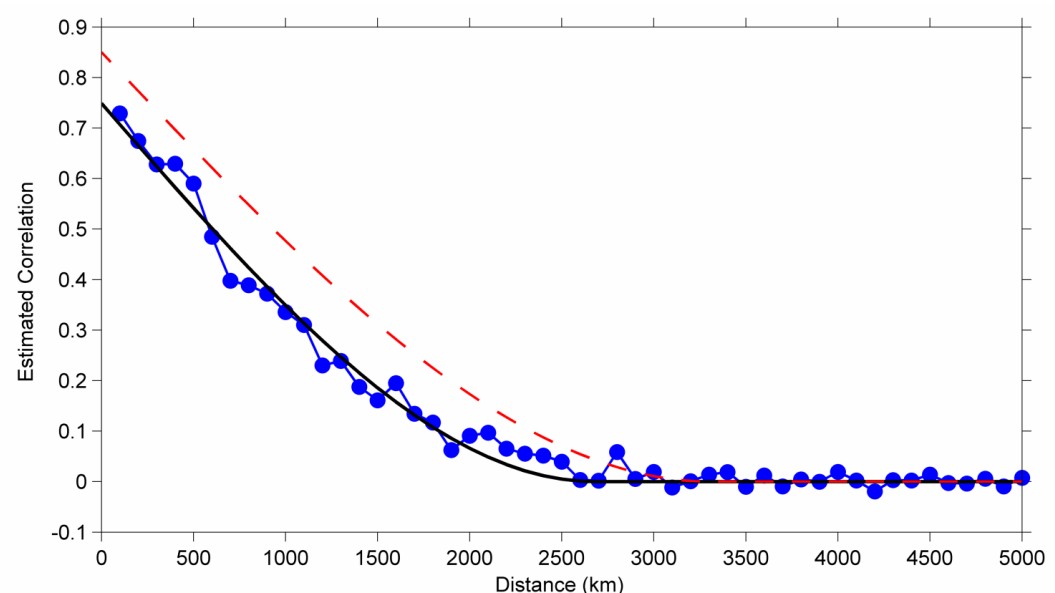

**Figure 1: Empirically estimated correlation versus distance for monthly average sea surface temperatures. Correlation was estimated by comparing root-mean-square differences for all possible pairs of HadSST grid cells and all months, and binning the population by distance. The black curve reflects a best fit for the spherical correlation function model. The red dashed curve shows the corresponding correlation model derived for land-based measurements (Rohde et al. 2013a).**

The distance correlation function gives rise to a Kriging formulation that:

$$T(x,t) = \theta_t + \sum_j \left( K(x_j, x, t)(SST(x_j, t) - \theta_t) \right) \qquad (2)$$

$$\begin{pmatrix} K(x_1, x, t) \\ \vdots \\ K(x_N, x, t) \end{pmatrix} = \begin{pmatrix} D(x_1, t) & R(\|x_1 - x_2\|) & \cdots & & R(\|x_1 - x_N\|) \\ R(\|x_2 - x_1\|) & D(x_2, t) & & & \vdots \\ \vdots & & \ddots & & \\ & & & D(x_{N-1}, t) & R(\|x_{N-1} - x_N\|) \\ R(\|x_N - x_1\|) & \cdots & & R(\|x_N - x_{N-1}\|) & D(x_N, t) \end{pmatrix}^{-1} \begin{pmatrix} R(\|x_1 - x\|) \\ \vdots \\ R(\|x_N - x\|) \end{pmatrix}$$

(3)

$$D(x_j, t) = \frac{1 + (N_{eff}(x_j, t) - 1)R_0}{N_{eff}(x_j, t)} \qquad (4)$$

$$N_{eff}(x_j, t) = \frac{s_m^2}{\left(\sigma_m(x_j, t)\right)^2}, \text{ minimum value of } 1 \qquad (5)$$

Where $t$ is the current month, $T(x, t)$ is the interpolated temperature at a general location $x$, $SST(x_j, t)$ is the HadSST anomaly value in the grid cell centered at location $x_j$, and $\sigma_m(x_j, t)$ is the measurement uncertainty associated with location $x_j$, and $s_m$ is the average measurement uncertainty of a single measurement. $N_{eff}(x_j, t)$ is then an effective number of independent measurements associated with the grid cell. Though HadSST provides the true number of observations per cell, $N(x_j, t)$, we found that $N_{eff}(x_j, t)$, which incorporates the measurement uncertainty appeared to give superior results than simply relying on the reported number of observations. The incorporation of $N_{eff}(x_j, t)$ into the determination of the Kriging coefficients $K$ has





the effect of giving greater weight to grid cells with less uncertainty. For integer values of $N_{eff}(x_j, t)$, the formulation of $D(x_j, t)$ is mathematically equivalent to having $x_j$ appear $N_{eff}(x_j, t)$ independent times in the correlation matrix. Note also that any empty HadSST grid cells at time $t$ are omitted from the matrix formulation for $K$.


$\theta_t$ is a free parameter at each time $t$ and effectively represents the global ocean average temperature anomaly. Its value is found iteratively by insisting that the spatial average of $T(x, t) - \theta_t = 0$.

It is instructive to note that this Kriging formulation has the property that $T(x_j, t) \rightarrow SST(x_j, t)$ in the limit that $N_{eff}(x_j, t) \rightarrow \infty$,

but will ordinarily produce a temperature estimate based on a weighted average of multiple HasSST grid points in the case that $N_{eff}(x_j, t)$ is small or moderate. The latter property can be useful in suppressing noise at grid locations with high uncertainty and/or very few measurements.

It is also important to recognize that that though the correlation function $R(d)$ has a very long tail, this does not mean that

average necessarily extends over a large area. In general, the Kriging coefficients $K(x_j, x, t)$ constructed in this way will heavily favor the nearest several data points. As long as nearby data is available, little weight will be given to distant grid cells. However, the long-tail of the correlation function means that the Kriging will attempt to fill large holes using distant data if no nearby data is available.

An absolute value field was also created by applying a similar interpolation to the HadSST climatology.

$$C(x, m) = P(x, m) + \sum_j \left( K_B(x_j, x, m)(SSTCLIM(x_j, m) - P(x, m)) \right) \qquad (6)$$

$C(x, m)$ is the interpolated climatology for month $m$, $SSTCLIM(x_j, m)$ is the reported climatology, $K_B(x_j, x, m)$ is a set of

Kriging parameters, which are the same as $K(x_j, x, m)$ except that $R_0$ and $D(x_j, t)$ are both replaced with 1, effectively treating the $SSTCLIM(x_j, m)$ as if it has no uncertainty. $P(x, m)$ a background prediction function dependent only on the month and the latitude of $x$. It is described as a piece-wise cubic spline with 11 knots as free parameters equally spaced in the cosine of latitude. These free parameters are chosen to minimize the spatial average of $C(x, m) - P(x, m)$. By construction, $C(x_j, m) = SSTCLIM(x_j, m)$ for all $x_j$, and this construction merely provides a way of interpolating between grid cell centers.


In addition to the above description, a physical cutoff was applied to the absolute temperature $C(x, m) + T(x, t)$ at a fixed minimum temperature of -1.8 C, which is freezing temperature of seawater. If the interpolation would suggest a value lower than this, $T(x, t)$ was adjusted accordingly to maintain the minimum value of -1.8 C. Such adjustments are rare.

Finally, one last interpolation is performed using an assumption of temporal persistence. Unlike land temperature anomalies, where the temporal correlation is often only a couple weeks, ocean temperature anomalies typically have a temporal correlation measured in months. This can be exploited to estimate ocean temperatures based on adjacent months when no other information is available.



Analogous to Rohde et al. 2013a, a diagnostic criterion can be constructed $V(x,t) = \sum_j K(x_j, x, t)$. Because of the nature of the Kriging coefficients, $V(x,t) \rightarrow 1$ in the presence of dense data and $V(x,t) \rightarrow 0$ if there is no HadSST data in the neighborhood of $x$.

The final estimate of the SST, including a temporal persistence adjustment for regions of low $V(x,t)$ is then


$$T_{final}(x,t) = T(x,t) + \left(1 - V(x,t)\right)\left(\frac{V(x,t+1)T(x,t+1) + V(x,t-1)T(x,t-1)}{V(x,t+1) + V(x,t-1)} - \theta_t\right) \qquad (7)$$

Here, t-1 and t+1 refer to the temperature field one month earlier and one month later, respectively. This adjustment allows for a modest reduction in uncertainty at early times when data is temporally sparse.


As described, this analysis is agnostic about the resolution used to sample the final temperature field. In practice, we generally use the same 15984-element equal-area grid as Rohde et al. 2013a to calculate $T_{final}(x,t)$, though with non-ocean elements masked out.

### 2.2 Ocean Uncertainty

The ocean-average uncertainty in our ocean reconstruction is estimated following essentially the same model as adopted by HadSST3. HadSST3 estimates the total reconstruction uncertainty as the combination of measurement uncertainty, coverage uncertainty, and bias uncertainty (Kennedy et al. 2011a, Kennedy et al. 2011b). Bias uncertainty, $\sigma_{bias}$, which reflects biases created due to variations over time in the ways that SST has been measured, is brought forward essentially unchanged by our analysis process (Figure 2). Due to its slowly varying nature, this uncertainty remains the most important limitation of the

detection of long-term averages.

The coverage uncertainty, $\sigma_{coverage}$, is the uncertainty in the large-scale average arising due to incomplete sampling of the spatial field. As with HadSST3, our estimate of the coverage uncertainty is constructed by sampling a known field, applying our interpolation procedure, and seeing how well we reproduce the underlying average of the known field. Following HadSST3, we

used the SST fields provided by HadISST v2 as our target. The HadISST fields are spatially complete, observation-based historical reconstructions of SST and sea ice concentration (Titchner and Rayner 2014). To estimate the coverage uncertainty associated with a specified HadSST sampling field, we mask every month of the HadISST dataset using that sampling field, interpolate the remaining data, and measure the error in the interpolated average relative to the true ocean-average of the whole HadISST field. The deviations in the ocean-average are then collected across all HadISST months and the uncertainty for that

coverage mask is reported as the root-mean-square average of the deviations. Using this technique, which is directly analogous to the HadSST3 coverage assessment technique, we estimate that the application of our interpolation approach typically reduces the coverage uncertainty by 20-40% (Figure 2).

Lastly, we consider the impact of our interpolation on the measurement and sampling uncertainty. Measurement uncertainty

essentially captures the errors in individual observations, while sampling uncertainty reflect the fact that water temperatures can





vary on timescales shorter than a month and spatial scales smaller than a grid box. Though interpolation does not change the underlying uncertainty associated with individual measurements, by adjusting the weight of individual observations in the overall average, we affect the way that individual measurement errors propagate into the global average. In particular, in the presence of sparse data, limited measurements may be extrapolated over a large area. In some circumstances, this can cause the effective

uncertainty in the global average due to these uncertainties to increase. In essence, the interpolation may trade improvements in coverage uncertainty against a greater impact for measurement uncertainty. This largely limits our ability to reduce the overall uncertainty by interpolation.

The impact of measurement uncertainty on a large-scale average depends on the error correlation. If the measurement

uncertainties were uncorrelated, then the error would generally be expected to decline with the square root of the number of measurements. In actuality, the measurement uncertainties are frequently correlated. In most cases, single ships report many measurements per month. Each of those measurements can have both random errors and a potential for systematic bias. For a single ship, we cannot expect this bias component of a measurement error to be reduced by increasing the number of observations. In their analysis HadSST3 models the entire error correlation matrix to understand the effect of measurement

errors on the global average uncertainty.

For HadSST3, the error correlation matrices were not published. As a result, it is not possible to exactly determine the effect of our interpolation procedure on the measurement uncertainty. However, we can make a reasonable estimate. Since HadSST3 releases both the per grid cell measurement uncertainties and the global average measurement uncertainty, we can compare the

expected measurement uncertainty treating all grid cell as independent to what is actually observed by HadSST3 using the whole error correlation matrix (Kennedy et al. 2011b).

$$\sigma_{uncorrelated} = \sqrt{\sum_j \left( A(x_j) \sigma_m(x_j, t) \right)^2} \qquad (8)$$

Where $A(x_j)$ is the fraction of the Earth's oceans represented by grid cell $x_j$ and $\sigma_{uncorrelated}$ is the measurement uncertainty resulting from assuming that the measurement errors in individual grid cells are uncorrelated with other grid cells.

We find that the measurement uncertainty reported by HadSST3 in the ocean-average is typically ~2.1 times larger than $\sigma_{uncorrelated}$, with some variation over time.


We use this estimate as a benchmark to approximate the effect of error correlation on our analysis of measurement uncertainty.

$$\sigma_{interpolated,\ measurement} = \frac{\sigma_{HadSST,\ measurement}}{\sigma_{uncorrelated}} \sqrt{\sum_j \left( \overline{K}(x_j, t) \sigma_m(x_j, t) \right)^2} \qquad (9)$$

$$\overline{K}(x_j, t) = \left( \iint K(x_j, x, t)\, dx \right) / \left( \iint 1\, dx \right) \qquad (10)$$


Where the double integral denotes the integral over the surface of the ocean. Thus $\overline{K}(x_j, t)$ is effectively the weight of the $x_j$ grid point in the global average.





The total uncertainty in the ocean-average is then found by assuming the components are independent.


$$\sqrt{\sigma_{bias}{}^2 + \sigma_{coverage}{}^2 + \sigma_{interpolated,\ measurement}{}^2} \tag{11}$$

In the early part of the time series, we find that interpolation does significantly reduce the uncertainty in the ocean-average. At late times, though coverage uncertainty is improved, bias uncertainty plays a large role and the total uncertainty in the ocean-235 average is little changed from the HadSST values. However, even if the ocean-average uncertainty is not changed, some users will nonetheless benefit from having a more spatially complete interpolated SST field.

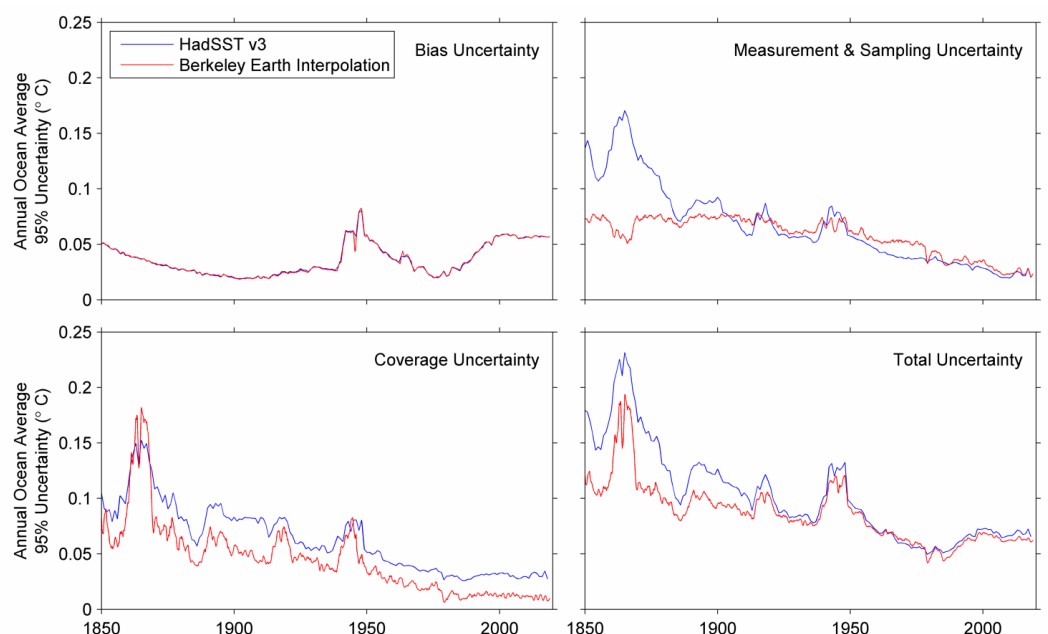

**Figure 2: Component uncertainties for the ocean-average of HadSST v3 and the corresponding transformed forms of those**
**components after the application of the interpolation scheme described in the text. All uncertainties are expressed as appropriate for**
**95% confidence intervals on annual ocean-averages.**

### 2.3 Land and Ocean Combination

The combined field is constructed by merging the Berkeley Earth Land Surface temperature with the interpolated SST field described above. Two versions are considered that differ only in their treatment of sea ice, using either the land air temperature
(LAT) or the SST field to estimate the temperature anomaly at sea ice locations. From 1850 to near-present, the sea ice locations are estimated using the ice concentration fields in HadISST v2 (Titchner and Rayner 2014).

To combine LAT and SST data, both data sets are expressed on the same grid. To simplify the combination at cells that are part-land and part-ocean, we have taken to adding in the spatial climatology and doing the combination in absolute temperatures.

In the case where sea ice areas are represented by SST, the combination is straightforward:





$$T_{combined}(x,t) = L(x)\, T_{LAT}(x,t) + (1 - L(x))\, T_{SST}(x,t) \qquad (12)$$

Where $L(x)$ is the fraction of the grid cell at location x that is land, and $T_{LAT}$ and $T_{SST}$ are respectively the LAT as estimated by Rohde et al. 2013a and the interpolated SST as described above.

In the case where sea ice regions are treated as land:

$$T_{combined}(x,t) = L^*(x,t)\, T_{LAT}(x,t) + \big(1 - L^*(x,t)\big)\, T_{SST}(x,t) \qquad (13)$$

$$L^*(x,t) = L(x) + \big(1 - L(x)\big)I(x,t) \qquad (14)$$

Where $I(x,t)$ is the ice fraction at location x at time t as reported by HadISST v2 (Titchner and Rayner 2014). For this purpose, HadISST is also regridded on to the same grid as LAT and SST. As HadISST is frequently delayed by a few months compared
to other climate data, it is necessary to supplement this data set when producing near real-time estimates. For this purpose, the Sea Ice Index of the National Snow and Ice Data Center (Fetterer et al. 2017) is used for months that are not yet available in HadISST. The modern ice distribution in both HadISST and the Sea Ice Index are based on satellite observations; however, we found that the Sea Ice Index tended to have systematically more partial melting than HadISST. To maintain consistency, a distribution transform was applied to the sea ice fractions provided in the Sea Ice Index based on comparing the 2014-2018 ice
fields in each dataset.

It is useful to note that regardless of whether one is using SST or LAT to estimate temperatures in association with sea ice, most such estimates involve a considerable extrapolation. In the case of LAT, for example, conditions over sea ice will usually be extrapolated from Greenland, Canada, Scandinavia and Russia. Whereas, when using SST, one extrapolates from rare SST
measurements that may be far removed from the sea ice edge. Or, in the case that analysis of the sea ice regions is excluded entirely, some methods are effectively substituting the ocean-average temperature anomaly.

It is our belief that the anomaly field generated by extrapolating air temperatures over sea ice locations is a more sensible approach to characterizing climate change at the poles. The air temperature changes over the sea ice can be quite large even
while the water temperatures underneath are not changing at all. In particular, over the last decades Arctic air has shown a very large warming trend during the winter.

Regardless of the approach used, the spatial climatology can then be calculated and removed (differing from the original only in cells with a mix of land and water/sea ice). Then the long-term trend in the climate can be computed using the spatial average of
the anomaly fields.

Uncertainties for the combined record are calculated by assuming the uncertainties in LAT and SST time series are independent and can be combined in proportion to the relative area of land and ocean. In the case that LAT is used over sea ice, the uncertainties for both LAT and SST have to be slightly recalculated by assuming that the time varying mask $L^*(x,t)$ is applied



the relevant spatial averages in the uncertainty estimations described in Rohde et al. 2013a and in the SST section above.  Doing

this adjustment causes a slightly increase in LAT uncertainty (due to the extrapolation over sea ice), and similar small decrease

in SST uncertanty.

## 3 Results and Conclusions

The global mean anomalies obtained from the Berkeley Earth land/ocean temperature record are quite similar to other published

records, as shown in Figure 3. With the exception of some short periods prior to 1880 and before and after World War 2, all four

other temperature records examined lie within the uncertainty envelope of the Berkeley Earth record. Differences around World

War 2 relate primarily to differences in adjustments to ERSST v4 and HadSST3 sea surface temperature records during that

period.

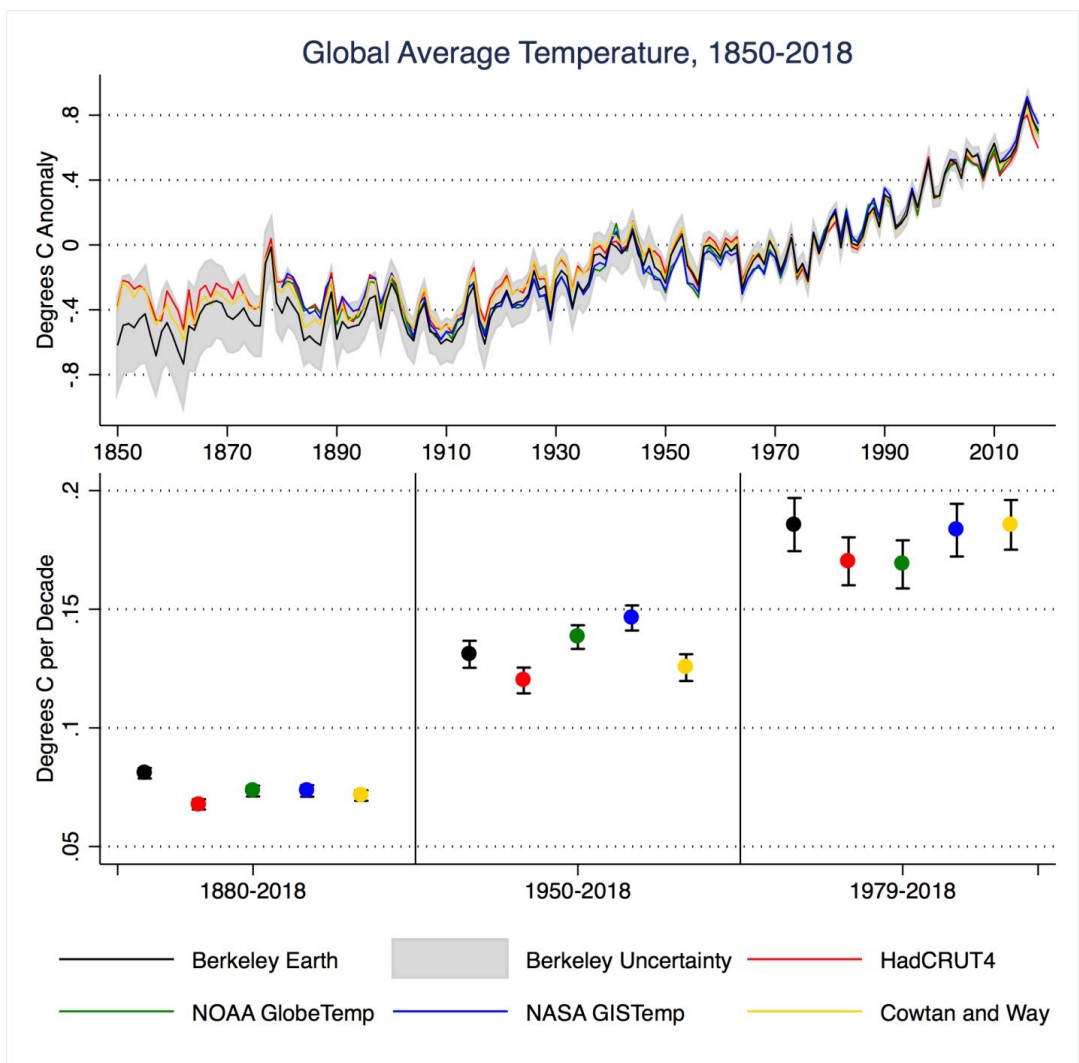


**Figure 3: Comparison of published global surface temperature records. The top panel shows annual anomalies (relative to a 1961-1990 baseline period), with the Berkeley Earth uncertainty as a shaded area. The bottom panel shows trends and two-sigma trend**



**uncertainties (calculated using an ARMA[1,1] approach to account for autocorrelation) for various starting dates through the end of 2015 based on monthly anomalies.**

Berkeley Earth has the highest trend of any temperature record examined for the period from 1880 to 2015, largely due to lower surface temperature estimates prior to 1900. These differences are driven both by increased spatial coverage from the inclusion of additional land records and the spatial interpolation of both land and ocean records (which is absent in NOAA and Hadley records). Similarly, Berkeley Earth has among the highest warming rates in the recent period (1979-2015) due primarily to greater Arctic coverage (where warming was unusually rapid during that period). The other record that provides robust arctic

interpolation, Cowtan and Way, also shows higher trends during this period.

From 1955 to present (after the availability of data in Antarctica), Berkeley Earth provides globally complete coverage via spatial interpolation, similar to NASA's GISTEMP and Cowtan and Way. This contrasts with HadCRUT4 and NOAA GlobalTemp which exclude any grid cells lacking station coverage or SST measurements. As shown in Figure 2, the patterns of

spatial anomalies between the different groups tend to be quite similar, apart from differences due to spatial coverage or gridded field resolution.

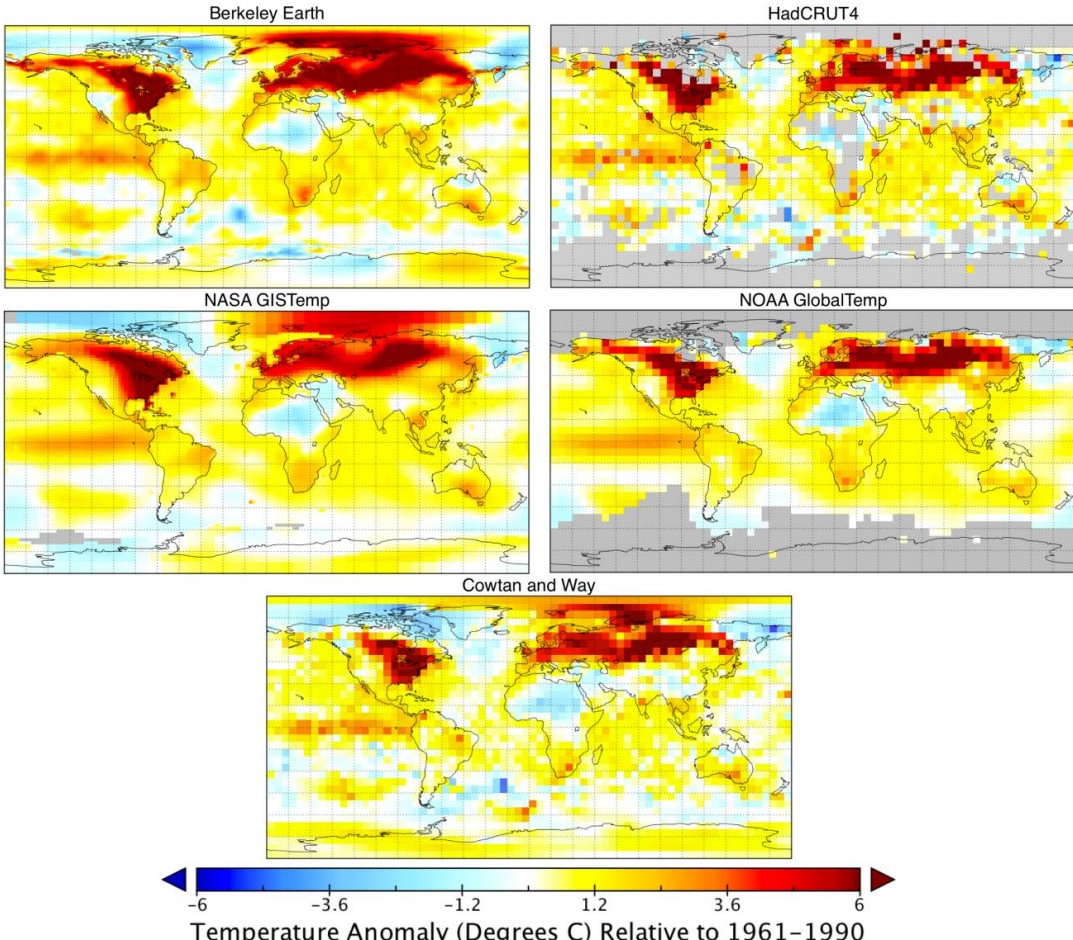

**Figure 4: Global gridded temperature anomalies for December 2015 relative to a 1961-1990 baseline for each global temperature dataset. Grid resolution is based on the highest resolution dataset provided by each group; 1x1 lat/lon for Berkeley Earth, 5x5 for HadCRUT4, 1x1 for NASA GISTEMP, 5x5 over land and 2x2 over oceans for NOAA GlobalTemp, and 5x5 for Cowtan and Way.**



When constructing a global surface temperature record, sea ice produces a challenging edge-case. The water temperature under sea ice is tightly constrained by the freezing point of water, and can only change with changes in sea ice cover. Air temperatures over sea ice are less well constrained, and can vary significantly over time. Whether areas with sea ice coverage are estimated

using sea surface temperatures or surface air temperatures will have a notable result on the record. While most groups (GISTEMP, Cowtan and Way) that interpolate temperatures over areas with sea ice cover use air temperatures, Berkeley Earth has provided both variants to allow researchers to select the series that best supports their needs. Both variants of the Berkeley Earth record are shown in Figure 5 as well as the HadCRUT temperature series for comparison; the use of SSTs under sea ice leads to lower warming trends in recent years, as it excludes air temperatures in parts of the Arctic that have been warming

rapidly over the past two decades.

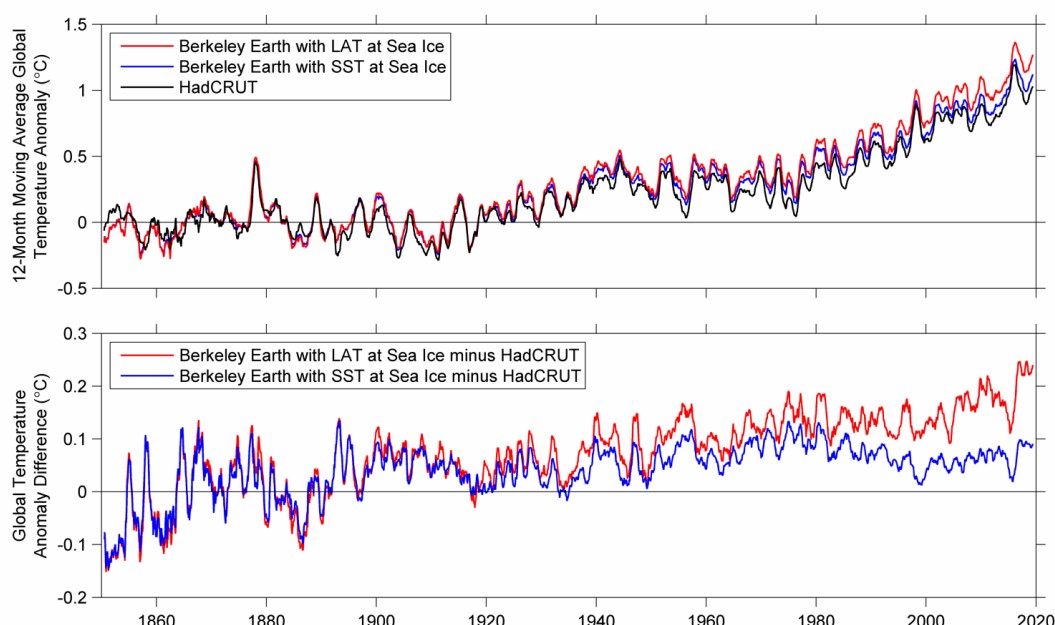

**Figure 5: (Top) Two variants of the Berkeley Earth global surface temperature product estimating temperatures under sea ice based on SSTs (red) or proximate air temperature measurements (blue), as well as the HadCRUT temperature series for comparison.**
**(Bottom) The same two versions of the Berkeley Earth data set with the HadCRU time series subtracted.**

Figure 5 also aids in understanding the difference between Berkeley Earth and HadCRUT. The interpolated SST field adopted here has a nearly identical trend to the HadSST field, differing by less than 0.01 °C / century. Part of the difference between Berkeley Earth's global temperature series and HadCRU is due to differences in the amount of warming estimated ot have occurred over land. This is the primary source of difference when comparing the Berkeley Earth series with SST at sea ice to the

HadCRUT series (blue line in Figure 5). While this difference is not insignificant, the larger difference is due to the treatment of the Arctic and the extrapolation of land temperature over sea ice areas (red line in Figure 5). Inclusion of the rapid warming above Arctic sea ice suggests the global average has increased an additional ~0.1 C during the last 100 years compared to estimates that do not include the changes in this region.

In addition to monthly temperature anomalies, Berkeley Earth produces monthly absolute temperature fields. A climatology field is estimated via Kriging observations, using elevation as a factor in the kriging process over land. Both absolute temperature



variants with air temperature over sea ice and water temperature under sea ice are available, as shown in Figure 4. Absolute temperatures are created by adding the climatology field to monthly anomalies.

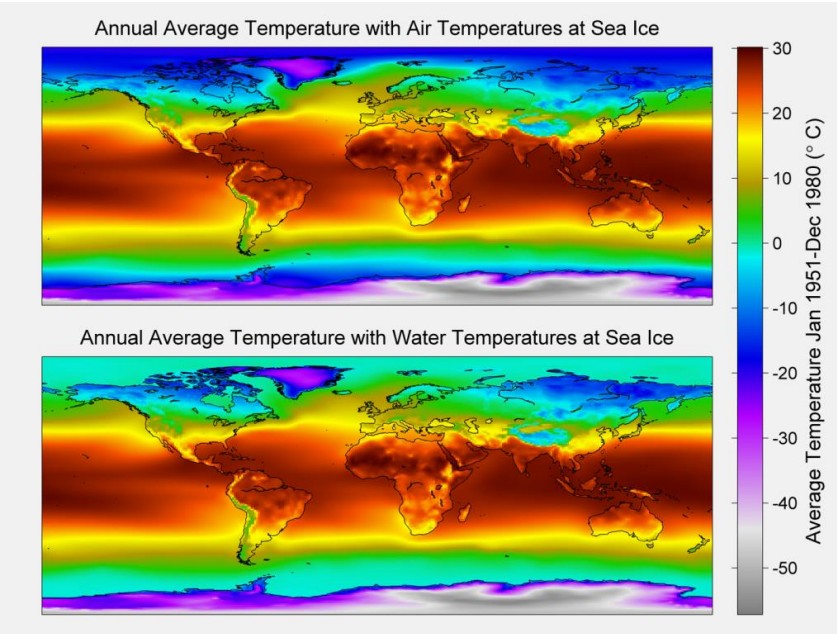


**Figure 6: Berkeley Earth average absolute climatology for the period from 1951-1980 with the air temperature at sea ice (top panel) and ocean temperature under sea ice (bottom) variants shown.**

Figure 7 provides a comparison between published uncertainties (two sigma) for each of the major global land/ocean temperature series. The Berkeley Earth, GISTEMP, and Cowtan and Way records have the lowest uncertainty of the groups providing annual

values, in part due to their spatial interpolation reducing the uncertainty associated with coverage.

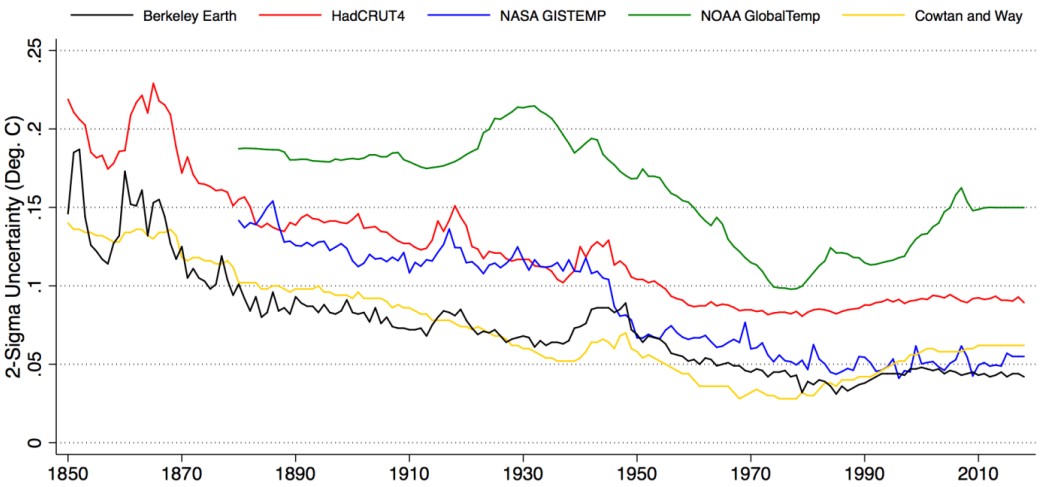

**Figure 7: Comparison of published annual uncertainty estimates (two sigma) for Berkeley Earth, HadCRUT4 (Morice et al 2012), GISTEMP (Lenssen et al 2019), GlobalTempv5 (Vose et al 2012), and Cowtan and Way (2014).**





The Berkeley Earth Land/Ocean surface temperature record presented here has already been used by a number of publications (e.g. Jones 2015; Thorne et al 2016; Sutton et al 2015). It joins a number of existing land/ocean surface temperature products that help provide a diverse examination of the Earth's changing climate since 1850, and can be used for diverse applications including climate model validation, estimating transient climate response, examining changes in extreme events, and other research areas.

**Data Availability**

The Land/Ocean temperature product will be updated monthly on the berkeleyearth.org website, and is freely available for use to all interested researchers. A convenience copy of the dataset available at the time this paper was created has been registered with Zenodo and is available at DOI:10.5281/zenodo.3634713 (Rohde & Hausfather 2020).

**Author contribution**

RR designed and implemented the data set construction. ZH provided feedback, graphics, and analysis. RR and ZH jointly prepared the manuscript.

**Competing interests**

The authors declare that they have no conflict of interest.

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
