# Peer review of "The Berkeley Earth Land/Ocean Temperature Record"

_Earth System Science Data, 2019_

## Short Comment (SC1) · 25 Jun 2020

Most of the cyclic temperature variation is due to ENSO and related climate indices. Could probably reconstruct most of the varying part of the temperature record with one time series selected from the equatorial Pacific. Save some time and effort
* * *
[Figure]

[Figure]

**Fig. 1.**

---

## Referee Comment (RC1) · Anonymous Referee #1 · 28 Jun 2020

**Review of "The Berkeley Earth Land/Ocean Temperature Record" by Robert A. Rohde and Zeke Hausfather**

The paper is a dataset paper describing a new merged global surface temperature product arising from a merge of the Berkeley Earth land product with the Hadley Centre's HadSST3 product. There is a clear need to document the merged product and the paper is well within the scope of the journal. The fundamentals of the method are sound. The paper thus should be published following revisions. I append comments differentiated by major and minor below for the authors to address.

**Major comments**

1. The biggest issue is not one of the authors own making, but rather indicative of renewed interests in surface temperatures and a proverbial race by dataset producers to create new and improved versions of products, presumably for inclusion in the upcoming IPCC assessment report. The implications are two fold for the present study as follows.

   Firstly, the Hadley Centre have recently updated their SST product to HadSSTv4 which builds upon the new ICOADS R3.0 which has considerably better coverage in several critical periods. It also does a better job of handling modern era biases. The dataset is available from https://www.metoffice.gov.uk/hadobs/hadsst4/ and my feeling is that it would future proof the current analysis to use the HadSSTv4 product rather than HadSST3 as the marine basis. I am assuming that updates to HadSSTv3 will stop when HadCRUTv5 becomes operational so this decision will be enforced onto the team sooner or later. Changing now would save the need for another paper / the situation where there is a mismatch between the paper and the operational product. Changing would also, presumably improve coverage in the historical eras and thus improve the analyses. Unless there is a compelling technical impediment to doing so I would urge the authors to switch over SST source to HadSSTv4 now.

   Secondly, the new versions of datasets and presence of new products means updates are likely warranted to the comparisons section. These include:
   - HadCRUTv5 (in final review) – contact Colin Morice for details
   - https://essd.copernicus.org/articles/11/1629/2019/essd-11-1629-2019.pdf - the Chinese merged product recently extended back to 1850
   - https://www.nature.com/articles/s41561-020-0582-5 which provides spatially complete estimation based upon HadCRUTv4
   - https://journals.ametsoc.org/bams/article/doi/10.1175/BAMS-D-19-0095.1/348446/The-EUSTACE-project-delivering-global-daily - new global surface air temperature estimates
   - https://journals.ametsoc.org/jcli/article/33/4/1351/346368/Uncertainty-Estimates-for-Sea-Surface-Temperature - substantially updated uncertainty estimates on the NOAA product

   Not all these need be used but clearly the HadCRUTv5 and updated NOAA estimates are key to include in revisions to at a minimum figure 7.

2. I am not entirely convinced there is merit in persisting with a version that uses SST under sea-ice as this clearly is not a surface temperature. The true surface temperature in such regions is either the ice (near-)skin temperature or the air temperature at some nominal height above the surface. There is a potential risk of mis-use of a product that considers sea surface temperature under ice as a reasonable estimate of a true surface temperature. It would possibly be better to discuss this but provide only the air temperature over ice version as this is the only realistic product over these regions. It would be good to back up with references the contention made line 277-281 in this regard.

3. The discussion in ln 30 to ln 34 is dated. The ICOADS release 3.0 is described in a newer manuscript (Freeman et al., 2017, doi:10.1002/joc.4775) and forms the basis for HadSSTv4 and ERSSTv5. The GHCN dataset has been updated in Menne et al., 2019 (https://doi.org/10.1175/JCLI-D-18-0094.1) and now includes very many more stations arising from the efforts of Rennie et al to improve land holdings. NASA don't, any longer, therefore use additional stations. This whole passage could be expanded and made a little more clear to provide a better and more accurate context for the reader here.

More generally the discussion of others efforts is somewhat perfunctory and in several aspects significantly dated. The whole introductory section requires substantive updates including several additional new products and new versions of products as noted in major comment 1. In particular, with the move to HadCRUT5 all products will employ some form of interpolation.

4. The temperature to distance correlation assumption in the ocean interpolation step is probably reasonable in ocean interior gyres. But it presumably breaks down in vicinity of upwelling, downwelling, coastal shelf seas and boundary currents. A little more justification / discussion is required than is given in ln 77-79. I suspect that you will need to provide a specific caveat about likely location-specific performance in such regions.

**Minor comments**

1. It feels dangerous to claim on ln 11 of the abstract that the product is more homogeneous. Without an absolute benchmark it is impossible, sadly, to say whether any given product is more homogeneous than another and aspects such as spatial smoothness can be misleading as pointed out in Sherwood et al., 2009 (https://rmets.onlinelibrary.wiley.com/doi/abs/10.1002/joc.1825). I don't see an explicit justification for such a statement from the underlying text.
2. There are several newer analyses than Cowtan et al. 2015 on ln 25 regarding SST/SAT. Richardson et al., 2018 plus newer in press papers by Lea Beusch, Nathan Gillett, Gareth Jones and others. It would also be worth being explicit how these measures may be expected to differ with time e.g. that SAT would be expected to warm / cool a little more than the underlying SST were to warm or cool.
3. Ln 28 the product is a joint effort between the Hadley Centre and UEA's CRU
4. Ln 28 Note newer references exist for the NOAA product and its uncertainty estimation (see major comments)

5. Please double check with GISS colleagues whether they continue to apply an additional night-lights based adjustment as my reading of Lenssen et al ended up ambiguous in this regard.

6. Both NOAA and NASA have switched to using ERSSTv5 so the text and reference ln 38-39 needs changing accordingly.

7. Globaltemp (ln40) does include some limited interpolation over land and is complete over the oceans. This needs to be corrected accordingly.

8. Ln 47-49 NOAA and NASA now use closer to 20 thousand stations following the GHCNv4 update and this should be reflected here.

9. Ln 233-236 makes little sense as written. I think you mean to say that in the more recent past coverage uncertainty diminishes in importance and bias uncertainty becomes increasingly important?

10. Ln 273-274 or Antarctic coastal stations, surely? There are two hemispheres with sea-ice …

11. Can a reference be given e.g. to the ERSSTv5 paper to back up the assertion on ln296-298? Note also that NASA and NOAA use ERSSTv5 and not v4 as noted in a prior comment.

12. In figure 3 are the trends OLS fits? You need to be explicit not just about the ARMA model applied but also the trend calculation method.

13. As noted in a prior comment NOAA is interpolated to an extent so Ln 307 and Ln313-314 require revision accordingly.

14. HadCRUT and not HadCRU in ln 338

---

## Referee Comment (RC2) · Anonymous Referee #2 · 8 Jul 2020

Review of "The Berkeley Earth Land/Ocean Temperature Record" by Robert A. Rohde and Zeke Hausfather

This paper describes the construction of the Berkeley Earth global temperature dataset, and is suitable for publication in this journal. I am happy to recommend publication and make some comments below.

Major comments:

1) As one of the other reviewers comments, the use of HadSST3 rather than HadSST4 is not ideal, although I appreciate that the HadSST3 version has been available for some time. I suspect the authors will also want to use HadSST4 (and will have to soon), but as long as the methods don't change when swapping for HadSST4 then this

paper sufficiently describes the approach to creating the overall dataset.

2) I also agree with one of the other reviewers when they question the use of a version of the dataset that uses SSTs under ice. Is this appropriate as a measure of global temperature change? But, if the authors want to include it then that should be their choice.

Minor comments:

1) Abstract: 'agrees quite well' is not very precise and could be expanded.

2) l23 - there are more observations of MATs than SSTs before around 1900 so this sentence could be edited slightly. There is a new NMAT dataset - Cornes et al. (in press) - for example.

3) l338 - typo 'ot' -> 'to'

4) Is it worth adding ERA5 to the post-1979 trends?

---

## Author Comment (AC1) · 25 Sep 2020

**1. The biggest issue is not one of the authors own making, but rather indicative of renewed interests in surface temperatures and a proverbial race by dataset producers to create new and improved versions of products, presumably for inclusion in the upcoming IPCC assessment report. The implications are two fold for the present study as follows.**

**Firstly, the Hadley Centre have recently updated their SST product to HadSSTv4 which builds upon the new ICOADS R3.0 which has considerably better coverage in several critical periods. It also does a better job of handling modern era biases. The dataset is available from https://www.metoffice.gov.uk/hadobs/hadsst4/ and my feeling is that it would future proof the current analysis to use the HadSSTv4 product rather than HadSST3 as the marine basis. I am assuming that updates to HadSSTv3 will stop when HadCRUTv5 becomes operational so this decision will be enforced onto the team sooner or later. Changing now would save the need for another paper / the situation where there is a mismatch between the paper and the operational product. Changing would also, presumably improve coverage in the historical eras and thus improve the analyses. Unless there is a compelling technical impediment to doing so I would urge the authors to switch over SST source to HadSSTv4 now.**

We are well aware of HadSSTv4 and are planning to incorporate it into the Berkeley Earth product when it is operational. Currently HadSSTv4 is not updated monthly (it only runs through December 2018), so it is not a good fit for a monthly-updated product like Berkeley Earth until it is made operational (which we suspect will occur in early 2021).

We have had conversations with John Kennedy about when this will occur and are preparing to make any needed tweaks based on (minor) changes in data format. Aside from certain minor changes in the communicated data format, HadSSTv4 can be used identically to HadSSTv3 in our analysis and should provide a drop-in replacement whenever the new version becomes operational.

[As an aside, we are also aware that HadCRUTv5 will be including a new interpolation of both CRUTEM and HadSST. In the future, we may choose to discontinue our own interpolation and replace it with the HadCRUTv5 approach. However, as the next version of HadCRUT is not yet published, we can't yet evaluate whether that course of action is desirable.]

We have updated the manuscript to read (lines 51-52):

"The ocean component of the land/ocean product uses an interpolated variant of HadSST v3, whose construction is described below. … Lastly, we note that HadSST v3 will be replaced with HadSST v4 once that product becomes operational (Kennedy et al. 2019). Aside from minor

differences in the way data is communicated and formatted, HadSST v4 should be usable following the same steps described here."

**Secondly, the new versions of datasets and presence of new products means updates are likely warranted to the comparisons section. These include: • HadCRUTv5 (in final review) – contact Colin Morice for details • https://essd.copernicus.org/articles/11/1629/2019/essd-11-1629-2019.pdf - the Chinese merged product recently extended back to 1850 • https://www.nature.com/articles/s41561-020-0582-5 which provides spatially complete estimation based upon HadCRUTv4 • https://journals.ametsoc.org/bams/article/doi/10.1175/BAMS-D-19-0095.1/348446/The-EUSTACE-project-delivering-global-daily - new global surface air temperature estimates • https://journals.ametsoc.org/jcli/article/33/4/1351/346368/UncertaintyEstimates-for-Sea-Surface-Temperature - substantially updated uncertainty estimates on the NOAA product Not all these need be used but clearly the HadCRUTv5 and updated NOAA estimates are key to include in revisions to at a minimum figure 7.**

While we agree with the reviewer's suggestions that HadCRUTv5 should be used if possible in Figures 3 and 7, we reached out to Colin Morice and he indicated that their paper will likely not be out before ours, and would prefer we not release their results before they do. We do not feel that excluding this yet-to-be-released dataset from our comparisons should be a reason to delay publication.

We have updated Figure 7 to include NOAA's revised uncertainty estimates. We have also changed references in the text from ERSSTv4 to ERSSTv5 to reflect the latest version of that dataset.

**2. I am not entirely convinced there is merit in persisting with a version that uses SST under sea-ice as this clearly is not a surface temperature. The true surface temperature in such regions is either the ice (near-)skin temperature or the air temperature at some nominal height above the surface. There is a potential risk of mis-use of a product that considers sea surface temperature under ice as a reasonable estimate of a true surface temperature. It would possibly be better to discuss this but provide only the air temperature over ice version as this is the only realistic product over these regions. It would be good to back up with references the contention made line 277-281 in this regard.**

We agree with the reviewer that the use of water temperatures in sea ice regions is not a natural way to think about global temperature changes, and we have now try to de-emphasize it for that reason.  However, we believe that it remains a useful comparison for a variety of reasons.

For one, ERSST (e.g. Huang et al. 2017) continues to have an adjustment that shifts SST estimates towards the freezing point of ocean water (-1.8 C) in regions of high sea ice concentration. Functionally, this is analogous to using ocean temperature in these sea ice regions. If one is specifically working with ERSST, then a treatment of SST under sea ice is a natural comparison.

However, one must also acknowledge that ERSST's adjustment has only a minimal impact on NOAA GlobalTemp since one of the final steps in constructing NOAA GlobalTemp is to mask all cells with a high sea ice concentration as missing (e.g. Vose et al. 2012).

Similarly, HadSST will only use SST values in partial sea ice regions (when data exists), though the availability of such measurements is relatively rare.

It is a limited audience, but people working specifically with ERSST or HadSST may naturally want to make comparisons with ocean temperature in the sea ice region.

A more important point is that this analysis allows us to call out the role of warming in sea ice areas on the overall global warming trend.

As the reviewer knows, there are important differences in how global averages consider the sea ice region. NASA GISTEMP uses air temperature extrapolation, similar to our preferred method. NOAA GlobalTemp simply omits missing values (including all regions with high sea ice concentrations) in performing their averages. Similarly, HadCRUT omits missing values in their averages, which will include most of the cells in sea ice areas. Omission of missing cells will in effect treat those regions as having an average equivalent to the global average of non-missing cells. Given the large role of Arctic amplification, omission of Arctic cells is likely to underestimate the true extent of global warming. While not exactly equivalent to omitting sea ice cells, the extrapolation of SST values into the sea ice regions also removes most of the effect of warming in these regions. Thus comparing the averages resulting from SST interpolation to that involving air temperatures helps to characterize the magnitude of the missing warming.

To summarize, we think the SST interpolation is somewhat informative and useful, though primarily of specialist interest. However, for most applications, the air temperature interpolation in the sea ice regions is the more applicable tool.

We have now added multiple comments to the paper to emphasize that the analysis using air temperatures over sea ice is the result that should be preferred in most applications.

**3. The discussion in ln 30 to ln 34 is dated. The ICOADS release 3.0 is described in a newer manuscript (Freeman et al., 2017, doi:10.1002/joc.4775) and forms the basis for HadSSTv4 and ERSSTv5. The GHCN dataset has been updated in Menne et al., 2019**

**(https://doi.org/10.1175/JCLI-D-18-0094.1) and now includes very many more stations arising from the efforts of Rennie et al to improve land holdings. NASA don't, any longer, therefore use additional stations. This whole passage could be expanded and made a little more clear to provide a better and more accurate context for the reader here. More generally the discussion of others efforts is somewhat perfunctory and in several aspects significantly dated. The whole introductory section requires substantive updates including several additional new products and new versions of products as noted in major comment 1. In particular, with the move to HadCRUT5 all products will employ some form of interpolation.**

We agree with the author that a number of these references are dated and reflect developments in the time between when this portion of the manuscript was originally written and ultimately submitted.

We have updated lines 26-43 to read:

"A number of prior groups have developed global land/ocean surface temperature indexes, including NASA's GISTEMP (Hansen et al., 2010; Lenssen et al 2019), Hadley/UEA's HadCRUT4 (Morice et al., 2012), NOAA's GlobalTemp (Smith et al., 2008; Vose et al., 2012; Huang et al 2020), and the Japan Meteorological Agency (JMA) (Ishihara 2006). Additionally, Cowtan and Way (2014) provide a spatially-interpolated variant of HadCRUT4 featuring greater spatial coverage. These series differ in a number of respects. They all largely utilize the same set SST measurements drawn from the ICOADS database (Freeman et al., 2017) and most of the same land temperature records contained in the Global Historical Climatological Network monthly database (GHCNm) (Lawrimore et al., 2011), though HadCRUT4 (and by extension Cowtan and Way) includes a more modest number of land stations than GISTEMP and GlobalTemp, which recently transitioned to using the much larger GHCNm v4 database (Menne et al 2018).

Both GISTEMP and GlobalTemp utilize NOAA's pairwise homogenization algorithm to detect and correct inhomogenities such as station moves or instrument changes in land stations (Menne and Williams 2009), though NASA applies an additional satellite nightlight-based urbanity correction (Hansen et al., 2010). GISTEMP and GlobalTemp both use NOAA's Extended Reconstruction Sea Surface Temperature (ERSST) version 5 (Huang et al., 2017) for SSTs, HadCRUT4 and Cowtan and Way use HadSST3 (Kennedy et al., 2011), and JMA uses COBE-SST (Ishii et al., 2005). HadCRUT4 and JMA include no spatial interpolation outside of 5-by-5 latitude/longitude grid cells, while GlobalTemp includes some interpolation over land but has nearly complete ocean temperature fields with the primary exception that sea ice regions are masked as missing.  GISTEMP and Cowtan and Way spatially interpolate temperatures out to regions with no direct station coverage (GISTEMP using a simple linear interpolation technique, while Cowtan and Way uses Kriging). The upcoming HadCRUT5 will transition to HadSST4 and include spatial interpolation (Morice et al, submitted)."

Menne et al 2018 GHCNv4 paper:
https://journals.ametsoc.org/jcli/article/31/24/9835/90961/The-Global-Historical-Climatology-Network-Monthly

Huang et al 2017 ERSSTv5 paper:
https://journals.ametsoc.org/jcli/article/30/20/8179/33181/Extended-Reconstructed-Sea-Surface-Temperature

Morice et al 2020, submitted.

**4. The temperature to distance correlation assumption in the ocean interpolation step is probably reasonable in ocean interior gyres. But it presumably breaks down in vicinity of upwelling, downwelling, coastal shelf seas and boundary currents. A little more justification / discussion is required than is given in ln 77-79. I suspect that you will need to provide a specific caveat about likely location-specific performance in such regions.**

The reviewer is correct that the use of a uniform correlation structure is an imperfect assumption.

Central ocean basins will actually have a longer average correlation than indicated, with polar regions, geographically complex basins, and regions of variable currents having a shorter average correlation.

On the monthly time scale, interpolation across these areas may be less reliable, and we have added a caveat to this effect.

"This treatment, using a single scale length for the whole ocean, simplifies the analysis; however, it does ignore some of the real variations across the oceans.  For example, in regions with boundary currents, upwelling/downwelling, or complex ocean to land geographies, the scale length of monthly average temperature variations may be smaller than suggested here.  In practice, the 5x5 degree gridding of HadSST already precludes a detailed analysis of most small features.  The interpolation presented here primarily serves to improve the representation by smoothing over noise and filling gaps, but it won't necessarily capture the smallest features."

While not discussed in this paper, on longer timescales, higher resolution models seem to suggest that temperatures in areas of boundary currents still tend to evolve at a similar rate to larger regional trends (with a few exceptions).  So, while failure to resolve small scale processes does reduce the utility of our local fields, it might not matter very much in the estimation of long-term trends in most cases.

**Minor comments**
**1. It feels dangerous to claim on ln 11 of the abstract that the product is more homogeneous. Without an absolute benchmark it is impossible, sadly, to say whether**

**any given product is more homogeneous than another and aspects such as spatial smoothness can be misleading as pointed out in Sherwood et al., 2009 (https://rmets.onlinelibrary.wiley.com/doi/abs/10.1002/joc.1825). I don't see an explicit justification for such a statement from the underlying text.**

We agree that this language is too strong, and have revised the line to avoid the impression that the dataset is more spatially complete or homogenous than existing datasets:

"It agrees quite well with records from Hadley's HadCRUT4, NASA's GISTEMP, NOAA's GlobalTemp, and Cowtan and Way, and provides a spatially complete and homogeneous temperature field."

**2. There are several newer analyses than Cowtan et al. 2015 on ln 25 regarding SST/SAT. Richardson et al., 2018 plus newer in press papers by Lea Beusch, Nathan Gillett, Gareth Jones and others. It would also be worth being explicit how these measures may be expected to differ with time e.g. that SAT would be expected to warm / cool a little more than the underlying SST were to warm or cool.**

We have revised line 25 to read:

"..though it is only an imperfect proxy and may be subject to slightly slower warming rates than marine air temperatures in recent decades (Cowtan et al., 2015; Richardson et al 2016; Jones 2020)."

Richardson et al 2016: https://www.nature.com/articles/nclimate3066
Jones 2020: https://rmets.onlinelibrary.wiley.com/doi/abs/10.1002/qj.3871

**3. Ln 28 the product is a joint effort between the Hadley Centre and UEA's CRU**

We have revised this to read "Hadley/UEA's HadCRUT4".

**4. Ln 28 Note newer references exist for the NOAA product and its uncertainty estimation (see major comments)**

Updated to read "NOAA's GlobalTemp (Smith et al., 2008; Vose et al., 2012; Huang et al 2020)"

**5. Please double check with GISS colleagues whether they continue to apply an additional night-lights based adjustment as my reading of Lenssen et al ended up ambiguous in this regard.**

Our understanding is that night lights still play a minor role though it has been reduced with the introduction of the homogenized station data from NOAA.

**6. Both NOAA and NASA have switched to using ERSSTv5 so the text and reference ln 38-39 needs changing accordingly.**

ERSSTv4 had been changed to ERSSTv5 there and on line 297.

**7. Globaltemp (ln40) does include some limited interpolation over land and is complete over the oceans. This needs to be corrected accordingly.**

The reviewer's caveat is partially incorrect. ERSST is complete over the oceans, but GlobalTemp masks as missing all cells with a sea ice concentration over 50% (Vose et al. 2012) so GlobalTemp has missing cells in both land and ocean areas.

We've updated this to read: "HadCRUT4 and JMA include no spatial interpolation outside of 5-by-5 latitude/longitude grid cells, while GlobalTemp includes some interpolation over land but has nearly complete ocean temperature fields with the primary exception that sea ice regions are masked as missing."

**8. Ln 47-49 NOAA and NASA now use closer to 20 thousand stations following the GHCNv4 update and this should be reflected here.**

We've update this to include the proper GHCNv4 number. The sentence now reads:

"The land data utilizes significantly more land station data (over 40,000 stations) compared to the ~10,000 land stations used by some of the other groups (though GISTEMP and GlobalTemp have both recently updated their records to include a larger number of land stations, including more 20,000 sites in GHCNv4)."

**9. Ln 233-236 makes little sense as written. I think you mean to say that in the more recent past coverage uncertainty diminishes in importance and bias uncertainty becomes increasingly important?**

We've rewritten this paragraph to make the intention clearer.

"Over all time periods, we find that interpolation does reduce the uncertainty associated with missing coverage. In the early period, this results in an appreciable reduction in total uncertainty. However, the total uncertainty in the global average is little changed in the recent period. This is because bias and measurement uncertainties play a dominant role in the recent period, and the impact of these uncertainties on the global average are little changed as a result of the interpolation. However, even if the ocean-average uncertainty is not changed during the recent period, the interpolation may still aid in the interpretation of local to regional-scale features."

**10. Ln 273-274 or Antarctic coastal stations, surely? There are two hemispheres with sea-ice …**

We've added mention for Antarctica.

**11. Can a reference be given e.g. to the ERSSTv5 paper to back up the assertion on ln296-298? Note also that NASA and NOAA use ERSSTv5 and not v4 as noted in a prior comment.**
We have added a references to Huang et al 2017, Kennedy et al 2019, and Cowtan et al 2017, which examines these issue in depth:

Differences around World War 2 relate primarily to differences in adjustments to ERSST v5 and HadSST3 sea surface temperature records during that period (Huang et al 2017; Kennedy et al 2019; Cowtan et al 2017).

Huang et al 2017 ERSSTv5 paper:
https://journals.ametsoc.org/jcli/article/30/20/8179/33181/Extended-Reconstructed-Sea-Surface-Temperature

Kennedy et al 2019: https://www.metoffice.gov.uk/hadobs/hadsst4/HadSST4_accepted.pdf

Cowtan et al 2017: https://rmets.onlinelibrary.wiley.com/doi/abs/10.1002/qj.3235

**12. In figure 3 are the trends OLS fits? You need to be explicit not just about the ARMA model applied but also the trend calculation method.**

We've clarified the caption to indicate that:

"The bottom panel shows trends calculated via an OLS fit and two-sigma trend uncertainties (calculated using an ARMA[1,1] approach to account for autocorrelation) for various starting dates through the end of 2019 based on monthly anomalies."

Similar approaches (central trend estimate from OLS, uncertainties from ARMA model) have been used in other papers, e.g. Hausfather et al 2017:
https://advances.sciencemag.org/content/3/1/e1601207

**13. As noted in a prior comment NOAA is interpolated to an extent so Ln 307 and Ln313-314 require revision accordingly.**

Line 307 has been revised to "...(which are more limited in both the NOAA and Hadley records)."

Lines 313-314 have been revised to "This contrasts with HadCRUT4 which exclude any grid cells lacking station coverage or SST measurements, or and NOAA GlobalTemp where interpolation is more limited."

**14. HadCRUT and not HadCRU in ln 338**

Fixed.

Reviewer 2

**1) As one of the other reviewers comments, the use of HadSST3 rather than HadSST4 is not ideal, although I appreciate that the HadSST3 version has been available for some time. I suspect the authors will also want to use HadSST4 (and will have to soon), but as long as the methods don't change when swapping for HadSST4 then this paper sufficiently describes the approach to creating the overall dataset.**

As discussed in our response to Reviewer 1, we are preparing to switch to HadSST4 once that dataset is operational and updated monthly (likely early in 2021). We have updated the manuscript to read (lines 51-52):

"Lastly, we note that HadSST v3 will be replaced with HadSST v4 once that product becomes operational (Kennedy et al. 2019). Aside from minor differences in the way data is communicated and formatted, HadSST v4 should be usable following the same steps described here."

**2) I also agree with one of the other reviewers when they question the use of a version of the dataset that uses SSTs under ice. Is this appropriate as a measure of global temperature change? But, if the authors want to include it then that should be their Choice.**

Please see our response to the other reviewer.  While we agree that extending the SST field under the sea ice is not a good measure of global temperature change, we do think it has some utility for making comparisons to other datasets that do this (e.g. ERSST) and for helping to characterize the effects of Arctic Amplification.

**Minor comments:**
**1) Abstract: 'agrees quite well' is not very precise and could be expanded.**

These comparisons are expanded later in the text, but we have changed it to "It provides a global mean temperature record quite similar to records from…" to be more precise in the abstract.

**2) l23 - there are more observations of MATs than SSTs before around 1900 so this sentence could be edited slightly. There is a new NMAT dataset - Cornes et al. (in press) - for example.**

We have changed it to "Sea surface temperatures are used in lieu of marine air temperatures due to relative scarcity post-1900 and…" to reflect this.

**3) l338 - typo 'ot' -> 'to'**

Fixed.

**4) Is it worth adding ERA5 to the post-1979 trends?**

While ERA5 is quite a useful dataset, it is substantially different in its construction from instrumental surface temperature records that we think its inclusion here is probably not warranted (though comparisons are certainly useful in other contexts!). The long length of other records would also make ERA5 difficult to see in the top panel of Figure 3.